

# Testing the enemy release hypothesis in a native insect species with an expanding range

Julia J. Mlynarek

Biology Department, University of Biology, Fredericton, New Brunswick, Canada

## ABSTRACT

The enemy release hypothesis (ERH) predicts that the spread of (invasive) species will be facilitated by release from their enemies as they occupy new areas. However, the ERH is rarely tested on native (non-invasive, long established) species with expanding or shifting ranges. I tested the ERH for a native damselfly (*Enallagma clausum*) whose range has recently expanded in western Canada, with respect to its water mite and gregarine parasites. Parasitism levels (prevalence and intensity) were also compared between *E. clausum* and a closely related species, *Enallagma boreale*, which has long been established in the study region and whose range is not shifting. A total of 1,150 damselflies were collected at three 'old' sites for *E. clausum* in Saskatchewan, and three 'new' sites in Alberta. A little more than a quarter of the damselflies collected were parasitized with, on average, 18 water mite individuals, and 20% were parasitized by, on average, 10 gregarine individuals. I assessed whether the differences between levels of infection (prevalence and intensity) were due to site type or host species. The ERH was not supported: *Enallagma clausum* has higher or the same levels of parasitism in new sites than old sites. However, *E. boreale* seems to be benefitting from the recent range expansion of a native, closely related species through ecological release from its parasites because the parasites may be choosing to infest the novel, potentially naïve, host instead of the well-established host.

## INTRODUCTION

The Enemy Release Hypothesis (ERH) predicts that a species will be successful in a new habitat when its former enemies (e.g., parasites) are not present (*Keane & Crawley, 2002*). This hypothesis has been widely applied to invasive species, such as plant or animal pests in new habitats (*Colautti et al., 2004*). There is mixed support for this hypothesis (*Heger & Jeschke, 2014*). A review by *Colautti et al. (2004)* found that the ERH was supported in 11 of 13 such studies. However, in comparisons of co-occurring evolutionarily related species, where one was an established resident and the other a newcomer, the ERH was supported in only a third of the studies. The reason for this lack of support for the ERH could be that the established species has evolved defenses against the enemy whereas the new species is still naïve and susceptible (*Colautti et al., 2004*). In other words, the invading species is joining a community that already has a close relative to which enemies

Submitted **8 July 2015**
Accepted **29 October 2015**
Published **19 November 2015**

Corresponding author
Julia J. Mlynarek,
julia.mlynarek@gmail.com

exist and so has enemies poised for potential attack on the invader, as seen in Darwin's Naturalization Hypothesis (*Darwin, 1859*; *Daehler, 2001*) or the Parasite Mediated Competition Hypothesis (*Price, Westoby & Rice, 1988*).

Since the review of *Colautti et al. (2004)*, the ERH has continued to be debated: the hypothesis was not supported in bullfrogs (*Dare & Forbes, 2013*) and gobies (Gobiidae) (*Kvach et al., 2014*) or leafrollers (Totricidae) (*Buergi & Mills, 2014*) but was demonstarted in ladybird beetles (Coccinellidae) (*Comont et al., 2014*) and brine shrimp (Artemiidae) (*Rode et al., 2013*). The ERH has been intensively studied with invasive species, particularly plants, where it has more support (*Liu & Stiling, 2006*). ERH has also been applied to host shifts (as enemy-free space), where the release from enemies comes not from removal in space but through movement to a species to a new host where enemies do not find that species (*Heard et al., 2006*). Other hypotheses, such as the Evolution of Increased Competitive Ability Hypothesis, have been proposed as alternatives. In this hypothesis, the invasive species evolve better competitive abilities than their close relatives over time (*Blossey & Notzold, 1995*). In a meta-analysis, there was little support for it (*Felker-Quinn, Schweitzer & Bailey, 2013*). However, studies testing these hypotheses have almost exclusively focused on invasive species.

Invasive species are not the only species whose ranges are changing; the ranges of native species are not static and change with climate or changing landscapes (*Chen et al., 2011*; *Burrows et al., 2014*). Shifts in species ranges can have consequences for species interactions, as a focal species encounters new partners or enemies and leaves old ones behind. Of course, range shifts can be caused by species interactions, as well; species can be pushed out of areas and into new ones by competition and predation, or can colonize new areas as they follow the range shifts or their own prey (*Davis et al., 1998*). However, there are few studies of species with shifting and expanding ranges (other than recently introduced species) that test and show support for the ERH. In a survey of common milkweed, *Asclepias syriaca* L. (Apocynaceae), natural populations at the geographical extremes of the ranges showed less herbivory (e.g., leaf damage and insect diversity) than those at the center of the range (*Woods et al., 2012*). *Prior & Hellmann (2013)* studied the oak-gall forming wasp *Neuroterus saltatorius* Hartig (Cynipidae), whose range is expanding northward, and found support for the ERH in both natural and experimental settings: the wasp had greater demographic success and fewer parasitoids in the new range. The ERH was also supported in a marine whelk, *Kelletia kelletii* (Forbes) (Buccinidae), with an expanding range (*Hopper et al., 2014*); despite poorer demographic performance in the expanded range populations, the whelk had only one fifth as many parasites as the historic-range populations.

I studied a native Nearctic damselfly, *Enallagma clausum* Morse (Odonata: Coenagrionidae), whose range is expanding westward in Canada. I tested the ERH by comparing prevalence and intensity of parasite infection in older, established populations of *E. clausum* (as a baseline) to those in newly occupied sites. I also contrasted parasitism of *E. clausum* with that of close relative, *Enallagma boreale* Selys (Odonata: Coenagrionidae), which co-occurs at both the old and new sites. *Enallagma clausum* is a widespread species

in Manitoba, Saskatchewan, and the northwestern United States (*Walker, 1953*), but has been moving westward into Alberta through the parklands ecoregion in the past few decades (*Acorn, 2004*). The reasons for expansion are unclear, but it is most likely due to increased availability of new habitat with increased standing water in the prairies during the summertime due to recently built cattle dugouts, irrigation canals, waterfowl projects, and power plants (*Acorn, 2004*). This westward expansion is being documented in many Lepidoptera as well as other odonates (J Acorn, pers. comm., 2011). *Enallagma boreale* is widespread throughout North America, its distribution reaching the Yukon, and has been established in Alberta for over 100 years (*Walker, 1953*).

The most common parasites of damselflies are water mites (Arthropoda: Acari) and gregarines (Protozoan: Apicomplexa) (*Corbet, 1999*). *Arrenurus* water mites (Arrenuridae) are external parasites of many insects associated with aquatic habitats (*Smith, Cook & Smith, 2010*). *Arrenurus* spp. are phoretically associated with the larval damselfly host; water mites attach parasitically and start feeding only on adult damselflies (*Smith, Cook & Smith, 2010*). Water mites feed on their hosts until fully engorged, and then drop back into the water to continue their life cycle. Gregarines, in contrast, are internal gut parasites of arthropods (*Clopton, 2009*). Gregarines develop and mate in the host mid-gut and are released into the environment as cysts. Gregarines are probably acquired from the environment; cysts have been observed on legs of prey items (*Åbro, 1976*).

The main objective of this study was to test the ERH by observing whether *E. clausum* is as frequently attacked by external and internal parasites in the newly occupied sites as in the 'old' sites where it has been long established. Secondarily, by observing *E. boreale*, I could determine whether the arrival of *E. clausum* either dilutes the parasite pool and spares the well-established close relative (*Brown, McPeek & May, 2000*), or increases the parasite pool and increases parasitism on both host species. I could therefore determine if at 'new' sites there are parasite populations that are locally adapted to their familiar host but not to the newly arriving one. If *E. clausum* is released from its enemies, I expect this species would have lower measures of parasitism in the new sites and there would not be a difference in parasitism in *E. boreale* between old and new sites. However, if there is another pattern of infection, such as differences in levels of parasitism in *E. boreale* or *E. clausum* not being freed from its parasites in the new sites, parasite-mediated competition, one host species benefits indirectly from the presence of a second related host species because the parasite has an alternate host (*Price, Westoby & Rice, 1988*) could be occurring.

## METHODS

### Host and parasite sampling

Adult hosts (*Enallagma clausum* and *Enallgma boreale*) were collected at six sites in the parklands ecoregion of Canada (Fig. 1). Adults were collected because it is at the adult life history stage that, both, successful water mite and gregarine parasitism can be assessed.

Lakes are all in the same ecozone, similar latitude and any other global parameters. They were chosen because of their proximity, their environmental similarities, and because they provide suitable habitat for *Enallagma* damselflies: all lakes are shallow, slightly saline, with

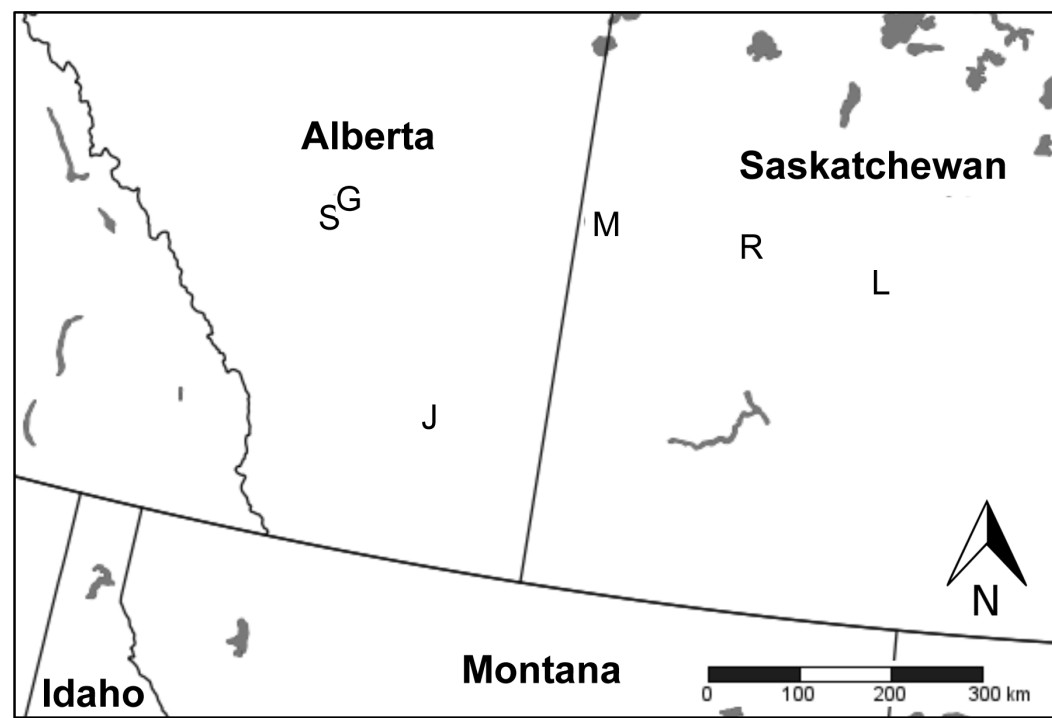

**Figure 1 Map of study sites testing the Enemy Release Hypothesis in a native species with an expanding range.** Map of study sites testing the enemy release hypothesis in the native species *Enallagma clausum* in Saskatchewan and Alberta. Old sites: L, Lenore lake; M, Manitou lake; and R, Redberry lake; New sites: G, Gull lake; J, Johnson lake; and S, Sylvan lake.

sandy bottoms and grassy vegetation along the edges. Three 'old' sites in Saskatchewan: Lenore Lake (52°28′56.86″N, 104°56′55.88″W), Manitou Lake (52°46′25.61″N, 109°44′19.37″W), and Redberry Lake (52°41′30.30″N, 107°10′17.71″W), were chosen because both damselfly species were known from these sites for at least 60 years (*Walker, 1953*). The old sites are an average distance of 200 km from each other. Three 'new' sites in Alberta: Gull Lake (52°29′28.87″N, 113°58′29.23″W), Sylvan Lake (52°20′42.34″N, 114°9′27.09″W) and Johnson Lake (50°35′44.60″N, 111°53′27.09″W), were selected based on *E. clausum* being first recorded in this area in the last decade (*Acorn, 2004*). Sylvan lake is 20 km from Gull lake and 240 km from Johnson lake. Although these lakes are similar ecologically, they range in size from one km$^2$ (Johnson Lake) to 80 km$^2$ (Gull Lake). In addition, unlike all the other lakes, Manitou is fishless. On a continental scale the lakes are close together, so there is the assumption that host phenological and ecological differences between the sites are negligible. No collecting permits were required for specimen collection.

Each site was visited twice in the first two weeks of July 2012, which is the documented peak time of activity for these two host species and their associations with parasites (*Walker, 1953*; *Mlynarek, Knee & Forbes, 2015*; *Mlynarek, Knee & Forbes, 2014*). During each site visit, 18–30 individuals of each damselfly species were collected by aerial sweep

net by a single collector walking through the grass along the shoreline of the sites. *Arrenurus* spp. infection was tallied in the field. A total of 497 *E. clausum* and 653 *E. boreale* hosts were examined for *Arrenurus* parasitism. Female damselflies were omitted from analyses because of low sample size and the potential for sex bias in parasitism (*Forbes & Robb, 2008*). A random subset of the damselfly specimens (342 *E. boreale* and 343 *E. clausum*) were collected and stored in separate vials in 95% ethanol for dissection in the lab to assess gregarine parasitism (see Table S1).

Water mites are usually located on either the ventral side of the thorax or the posterior ventral portion of the abdomen (*Smith, Cook & Smith, 2010*). All water mites were tallied and treated as a single taxonomic unit to compare prevalence and intensity of parasitism (*Corbet, 1999*; *Smith, Cook & Smith, 2010*) because the mites are largely generalist species that would be expected to have similar host ranges (*Mlynarek et al., 2015*).

To determine gregarine infection, damselfly abdomens were dissected by tearing the membrane between the dorsal tergites and the ventral sclerites to expose the gut. Once the gut was exposed the gregarine individuals could be tallied under $10\times$ magnification. As with water mites, all gregarines were combined as one taxonomic unit based on the assumption that gregarines are generalists and are expected to infect evolutionary congeners.

## Analyses

Prevalence (proportion of infected individuals per site) and mean intensity (average number of parasites per infected individual) were measured for each parasite group. Confidence intervals for both measures were calculated using QP3.0 (*Rózsa, Reiczigel & Majoros, 2000*).

Two-way ANOVAs were used to describe the direct and interactive effects of 'species' and 'site type' ('old' vs. 'new') on the prevalence and intensity of parasitism by water mites and gregarines. Four two-way ANOVAs were performed, one for each measure of parasitism and for each parasite group. All analyses were performed in JMP v.11 (SAS 2013).

## RESULTS

### Water mite parasitism

Among sites, water mite prevalence on *E. clausum* varied between 0.05 ([0.01–0.1] 95% Clopper–Pearson confidence interval) and 0.71 ([0.58–0.82] 95% CI; Table 1). Intensity varied between 12.8 ([3.07–31.14] 95% bootstrap CI) and 36.8 mites per infected host ([19.8–50.7] 95% bootstrap CI; Table 1). In *E. boreale*, prevalence varied between 0.11 ([0.06–0.16] 95% CI) and 0.65 ([0.49–0.78] 95% CI). Intensity varied between 8.05 ([5.27–12.6] 95% bootstrap CI) and 27.9 ([20.8–37.1] 95 % bootstrap CI; Table 1).

There were significant differences in *Arrenurus* prevalence in the species by site interaction ($F_{1,11} = 22.2$; $P < 0.01$; Table 2A). *Enallagma clausum* had a significantly higher prevalence of *Arrenurus* parasites at new sites than at old sites (Fig. 2A) whereas, *Enallagma boreale* showed a higher prevalence of *Arrenurus* parasites at old sites (Fig. 2A). There was a significant species $\times$ site type interaction for *Arrenurus*

**Table 1 Prevalence and intensity of *Arrenurus* water mite and gregarines on *Enallagma boreale* and *Enallagma clausum* from six sites in Eastern Alberta and Western Saskatchewan (see Table S1 for raw data).** Prevalence with Clopper Pearson 95% confidence limits and mean intensity with Bootstrap (BCa) 95% confidence limits with 2,000 replications.

| Species | Site type | Site | *Arrenurus* | | | Gregarine | | |
|---|---|---|---|---|---|---|---|---|
| | | | N | Prevalence | Intensity | N | Prevalence | Intensity |
| *E. boreale* | Old | Lenore | 61 | 0.32 (0.24–0.49) | 9.00 (5.27–12.64) | 51 | 0.36 (0.26–0.54) | 10.59 (6.05–21.20) |
| | | Manitou | 186 | 0.37 (0.31–0.46) | 8.59 (6.24–12.27) | 69 | 0.31 (0.22–0.46) | 11.80 (8.26–18.48) |
| | | Redberry | 48 | 0.51 (0.49–0.78) | 27.48 (20.81–37.10) | 43 | 0.22 (0.15–0.44) | 9.67 (4.83–16.50) |
| | New | Gull | 131 | 0.18 (0.11–0.25) | 7.92 (3.70–19.22) | 65 | 0.09 (0.03–0.19) | 7.00 (2.00–13.50) |
| | | Johnson | 59 | 0.24 (0.14–0.37) | 17.47 (9.64–36.14) | 59 | 0.45 (0.34–0.61) | 9.79 (5.96–14.93) |
| | | Sylvan | 168 | 0.11 (0.06–0.16) | 22.42 (14.72–41.78) | 55 | 0.12 (0.05–0.24) | 21.86 (7.00–46.14) |
| *E. clausum* | Old | Lenore | 67 | 0.08 (0.01–0.13) | 10.67 (2.00–29.33) | 48 | 0.09 (0.03–0.23) | 9.00 (3.60–17.00) |
| | | Manitou | 38 | 0.06 (0.01–0.18) | 16.67 (1.00–24.50) | 37 | 0.08 (0.01–0.18) | 2.25 (2.00–4.50) |
| | | Redberry | 174 | 0.05 (0.02–0.10) | 38.45 (19.76–50.67) | 70 | 0.03 (0.00–0.08) | 17.50 (N/A) |
| | New | Gull | 83 | 0.17 (0.1–0.27) | 20.00 (3.07–31.14) | 65 | 0.07 (0.02–0.15) | 3.80 (2.00–6.50) |
| | | Johnson | 73 | 0.29 (0.19–0.41) | 13.77 (7.96–23.19) | 61 | 0.42 (0.29–0.54) | 11.41 (7.80–16.48) |
| | | Sylvan | 62 | 0.67 (0.58–0.82) | 26.98 (20.39–36.41) | 62 | 0.11 (0.06–0.24) | 5.38 (2.88–9.50) |

intensity as well ($F_{1,11} = 6.16$; $P = 0.02$; Table 2B). Again, intensity was higher for *E. clausum* at new sites then old sites, but lower at new sites then old sites for *E. boreale* (Fig. 2B).

## Gregarine parasitism

Gregarine prevalence in *E. clausum* varied between 0.01 ([0.00–0.08] 95% CI) and 0.41 ([0.29–0.54] 95% CI; Table 1) over all the sites. Intensity varied between 3.50 ([2.00–3.50] 95% bootstrap CI) to 11.56 ([7.80–16.48] 95% bootstrap CI; Table 1). In *E. boreale*, gregarine prevalence varied between 0.09 ([0.03–0.19] 95% CI) and 0.48 ([0.34–0.61] 95% CI; Table 1). Intensity varied between 7.00 ([2.00–13.50] 95% bootstrap CI) and 22.43 ([7.00–46.14] 95% bootstrap CI; Table 1).

There were non-significant differences in species by site type for gregarine prevalence ($F_{1,11} = 3.68$, $P = 0.07$; Table 3A), or in species by site type for gregarine intensity ($F_{1,11} = 2.47$, $P = 0.13$; Table 3B). However, the gregarine prevalence was close to
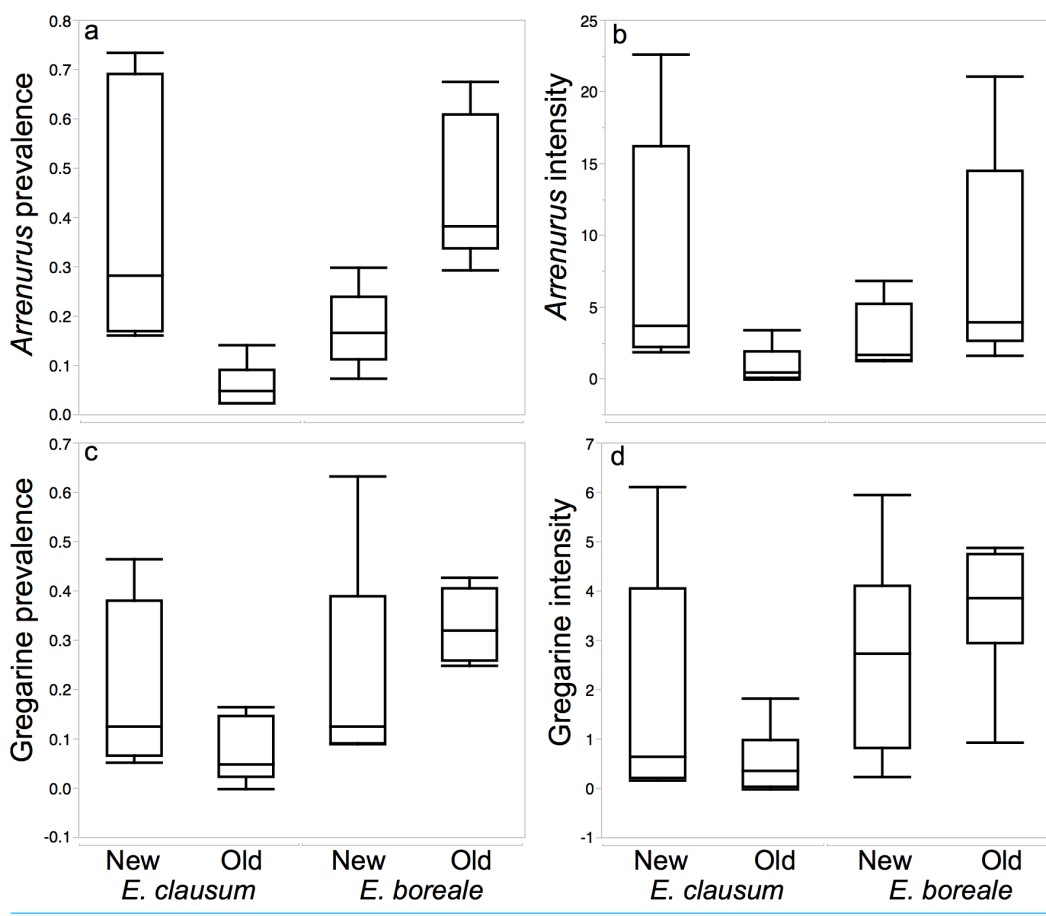

**Figure 2 Boxplots demonstrating differences in measures of parasitism between host species and sites types while testing for the Enemy Release Hypothesis in a native damselfly species with an expanding range.** Boxplot of differences in (A) *Arrenurus* prevalence, (B) *Arrenurus* intensity, (C) gregarine prevalence and (D) gregarine intensity between old and new sites for *Enallagma clausum*, a native species whose range is expanding, and a closely related, well-established species, *Enallagma boreale*.

**Table 2 Results of two-way ANOVA testing for differences between host species and site (Old vs. New) for (A) *Arrenurus* prevalence and (B) *Arrenurus* intensity infections in *Enallagma clausum*, a native species whose range is expanding in western Canada, and *Enallagma boreale*, a closely related long-established species.**

| Source | df | SS | F | P |
|---|---|---|---|---|
| (A) Prevalence | | | | |
| Species | 1,11 | 0.05 | 1.93 | 0.18 |
| Site | 1,11 | 0 | 0.18 | 0.66 |
| Species*site | 1,11 | 0.53 | 22.18 | **<0.01** |
| (B) Intensity | | | | |
| Species | 1,11 | 3.51 | 0.1 | 0.75 |
| Site | 1,11 | 8.42 | 0.25 | 0.62 |
| Species*site | 1,11 | 209.8 | 6.16 | **0.02** |

**Table 3** Results of two-way ANOVA testing for differences between host species and site (Old vs. New) for (A) gregarine prevalence and (B) gregarine intensity infections in *Enallagma clausum*, a native species whose range is expanding in western Canada, and *Enallagma boreale*, a closely related long-established species.

| Source | df | SS | F | P |
|---|---|---|---|---|
| (A) Prevalence | | | | |
| Species | 1,11 | 0.12 | 5.89 | 0.03 |
| Site | 1,11 | 0 | 0.05 | 0.83 |
| Species*site | 1,11 | 0.08 | 3.68 | 0.07 |
| (B) Intensity | | | | |
| Species | 1,11 | 23.1 | 7.41 | 0.01 |
| Site | 1,11 | 0.21 | 0.07 | 0.8 |
| Species*Site | 1,11 | 7.69 | 2.47 | 0.13 |

significant, paralleling the pattern of *Arrenurus* mites where gregarine prevalence tended to be higher at new sites for *E. clausum* and at old sites for *E. boreale* (Figs. 2C and 2D).

## DISCUSSION

The Enemy Release Hypothesis (ERH) predicts that a species whose range is expanding should have lower levels of parasitism in 'new' sites *versus* 'old' sites. In this case, *E. clausum* had significantly higher prevalence and intensity of *Arrenurus* in 'new' sites than in 'old' sites. Gregarine prevalence was also greater in new sites compared to old sites, though only to a marginally significant degree. There was no significant difference in gregarine intensity. Another expectation of the ERH is that a recently colonized species should have lower levels of parasitism than a long-time resident in the same new geographic area. This effect was not observed in either measure of water mite or gregarine parasitism. Instead, there was significantly higher *Arrenurus* parasitism in the newly occupied sites in *E. clausum*, the species whose range is expanding, compared to both *E. boreale* and to *E. clausum* populations at old sites. There could be a confounding effect with the pooling of species of each parasite group together to test ERH on the prevalence and intensity of parasitism rather then on specific parasite identity. This pooling of each parasite group into a morpho-group may mask what is happening at the species level. There is a possibility that different species might adapt differently to different hosts, and may be at different densities and thereby creating confounding problems when pooled. Because of the issues with parasite identification, the aim of this study to test ERH on particular parasite species but on the prevalence and intensity of morphogroups and could determine that there are clear differences in levels of parasitism. I can conclude that the ERH does not describe the patterns of either water mite or gregarine measures of parasitism (such as prevalence and intensity) on *E. clausum*.

An alternative explanation for these results involves parasite-mediated competition, where a host species benefits indirectly because of the presence of a new closely related host (*Price, Westoby & Rice, 1988*), or Darwin's Naturalisation Hypothesis, where the new species will not be at an evolutionary advantage because a close relative is already present

in those habitats and has enemies (*Daehler, 2001*). These two *Enallagma* host species are probably in competition because they are closely related (*Brown, McPeek & May, 2000*) and have similar ecological habits. There are some behavioural differences; in Eastern North America *E. boreale* develops in fish and *E. clausum* in dragonfly habitats (*McPeek, 1990*). In this study, both species were collected in sympatry, so it can be assumed that as larvae they developed in the same lake and were under the same pressures. *Enallagma boreale* may be better at evading these populations of parasites because it has had more time to coevolve with them at the sites than *E. clausum*. *Enallagma clausum* and *E. boreale* obviously share a close evolutionary history (*Brown, McPeek & May, 2000*) and can share parasites across sites (*Mlynarek, Knee & Forbes, 2015*), but *E. clausum* does not share an ecological evolutionary history with the particular parasite population at the new sites. The parasites have a naïve close relative to exploit (*Maron & Vila, 2001*) a pattern also observed in anolis lizards (*Schall, 1992*) and in oak herbivory, where there is increased herbivory on non-native oak species in the presence of native close relatives (*Pearse & Hipp, 2014*).

Another alternative for the observed patterns in *E. clausum* may involve the range edge effect. There are many contributing factors that limit the extent of a species range limit, one of which is increased parasite rates (*Sexton et al., 2009*). For example, populations of *Calopteryx maculata* at the northern edge of their Canadian range have higher levels of parasitism than those closer to the center of their range (J Mlynarek, 2011, unpublished data). Additionally, *Kaunisto & Suhonen (2013)* suggested that host damselflies at the edges of their ranges are under higher environmental stress, and therefore may be more susceptible to parasites and infections.

It is possible that *E. clausum* did undergo enemy release when it first arrived at the sites but the parasites subsequently quickly evolved to include them as a host. In this context, the data does not support the Evolution of Increased Competitive Ability Hypothesis, either. However, a decade is a short time period of time in an evolutionary context, considering that water mites and damselflies tend to have univoltine life cycles in these areas (*Smith, Cook & Smith, 2010*). *Enallagma clausum* has recently expanded its range on a continent it has long occupied. It may, therefore, not be leaving its enemies behind, in the sense that the mites and gregarines from its former geographic range may be only slightly different from those in its newly occupied range. Over the course of the ever-changing post-Pleistocene environment, with changes in both climate and the distribution of appropriate habitats, damselflies and their parasites have likely existed in a continually shifting dynamic for millennia. Indeed, species that are expanding their range may actually be under higher parasitism pressure in new sites, whereas those already present may benefit from their arrival in the short term.

## ACKNOWLEDGEMENTS

I thank John Acorn and Naomi Cappuccino for insights into this project, and Terry Wheeler and Stephen Heard for comments on an earlier draft.

### Funding

This work was funded by an Entomological Society of Canada Graduate Research Travel Grant and a Natural Sciences and Engineering Research Council of Canada Graduate Scholarship. The funders had no role in study design, data collection and analysis, decision to publish, or preparation of the manuscript.

### Grant Disclosures

The following grant information was disclosed by the author:
Entomological Society of Canada Graduate Research Travel Grant.
Natural Sciences and Engineering Research Council of Canada Graduate Scholarship.

### Competing Interests

The author declares there are no competing interests.

### Author Contributions

- Julia J. Mlynarek conceived and designed the experiments, performed the experiments, analyzed the data, contributed reagents/materials/analysis tools, wrote the paper, prepared figures and/or tables, reviewed drafts of the paper, collecting the data.

### Supplemental Information

Supplemental information for this article can be found online at http://dx.doi.org/10.7717/peerj.1415#supplemental-information.

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
