# Peer review of "Testing the enemy release hypothesis in a native insect species with an expanding range"

_PeerJ, doi:10.7717/peerj.1415_

## Round 0.1 · original submission · Major Revisions

· Academic Editor

Major Revisions

The question you are addressing is important and your overall study system is interesting. However, after reading the ms and the reviewers' comments carefully, I see two significant problems that you may not be able to address in a revised ms. Those are (1) the timing of sampling and (2) the host life-stage sampled. Both can significantly bias your findings and invalidate your conclusions. If you can address both of them fully please go ahead and submit a revised ms.

Reviewer 1 ·

Basic reporting

This manuscript is clearly written and addresses an interesting topic of range-expansion and enemy-release in insects, which are not as robustly evaluated for ERH as invasive plants. The background information provided is appropriate and well organized but could be flushed out a bit more. Additional information on behavior of parasitized damselflies would be useful as it could impact sampling effort and skew results. Parasitized insects have been shown to behave differently than their 'healthy' counterparts.It is also not stated what the dispersal range is of these damselflies. Do they not disperse from their natal habitat?

Experimental design

The experiment as laid out will test the stated hypotheses. It was not clear to me at first that the analysis was conducted on new vs. old lakes and should be clarified in the analysis section. Was site (Lake) not a significant factor?
A couple comments about the data analysis. Did the author analyze the data for assumptions of normality? As with most count data, there are a lot of zeros. Although this is taken into account somewhat with the way the data is analyzed (binomial evaluation of prevelence and intensity) it should be stated that the data meets this assumption, if it does. Since the author is analyzing proportional data (Prevalence) in an ANOVA it should be arcsin transformed. The information presented in Tables 2 and 3 have errors in the df values - they are reported as 1 and 11 in the manuscript but only as 1 in the tables.

The apparent differences presented in Fig 2 showing differences in parasitism between old and new lakes is really interesting and very relevant to the hypothesis on the adult stage.

The attached dataset has a lot of missing data values, including values for gregarine parasites.

Validity of the findings

The results are very interesting but need to be assessed again after data is analyzed with the appropriate transformations. As presented, the data well supports the authors conclusions that E. clausum adults do not experience ERH in newly invaded habitats. However, evaluating the adult stage is not an appropriate to test the ERH for this parasite-species complex and is a serious flaw in the validity of the experiment. As stated in the introduction, the parasites are phoretically acquired during the larval stages. Enemy release against the larval life stage would be a more appropriate test unless the authors are sure that the adults collected developed in the lake where they were collected. If the dispersal range of these damselflies is so short it should be clarified in the introduction.

·

Basic reporting

I found the writing a bit confusing and poorly constructed, in places. I have made many stylistic suggestions for the author. The final grammar and construction are the purview of the author, of course, but I encourage the author to consider my suggestions.

Experimental design

This study addresses a classic ecological hypothesis in a straightforward way, with an appropriate experiment and appropriate statistical analyses.

Validity of the findings

I have a couple more significant methodological points. First, each site was sampled only twice, and all sites were sampled during the same 2-week period in July. Given that the sites are so widely separated, isn’t it likely that the sites are at different points during the population cycles of the mites? The J site, for instance, is at much lower latitude and might be “further along into summer” than the other sites – perhaps reflected in the generally higher levels of parasitism at this site compared to the others. Likewise, are the three new sites in Alberta at higher elevations than those in Sask? It just seems that, since the sampling was done over such a short period but over such a large geographical area, that differences both between and within “site” categories could be affected by differences in phenology.
Second, pooling all mites and gregarines is a significant methodological choice that needs to be addressed. The author states that pooling was justified because species within a parasite type have the same effect on the host. Although they do, I don’t think this is the issue. Although two mite species both have the same physiological effect on a host (sucking hemolymph), it seems possible that the host may be able to mount an immunological response against one and not the other. It is possible that a naïve species might be susceptible to one resident mite species but not another, and these patterns are confounded by pooling all mite species. Pooling might be justified on other grounds: mites are typically generalist species, and so SHOULD be able to exploit closely related hosts. Multiple generalist mite species might be considered as an “ecological species” not because they have the same affect, but because they are generalists that should be able to exploit the same species. If they don’t, as a pooled group, then something else is happening – like a host being differentially susceptible to one or all of the mite species. Pooling parasites is a methodological necessity for studies conducted over several sites with large samples of hosts; I have done it, myself. But while it is a necessary convenience, the potential confounding effects of this choice should be considered explicitly… especially given the POPULATIONAL specificity that the author is trying to test here. The ERH assumes that populations of the same parasites will have adapted differently to different hosts. Surely the possibility that different species might adapt differently to different hosts, and might be at different densities in such widely spaced sites, and thereby create confounding problems when pooled, needs to be addressed as a caveat in the discussion.

Additional comments

This is an appropriate study of an ecologically interesting hypothesis. The methods and analyses are sound, but the short sampling window and the pooling of parasites create some possible confounding effects that need to be alluded to, at least. But this is a valuable large-scale survey of parasitism in two congeners.

---

## Round 0.2 · accepted · Accept

· Academic Editor

Accept

I was asked to take over as Academic Editor of this MS because the previous Academic Editor has become unreachable for reasons unknown.

Having read the previous reviews, the revised MS, and the second reviews, I agree with the reviewers that this MS should now be accepted for publication in PeerJ. The results represent an interesting piece of natural history and add to the literature on the ERH.

I did note a few small typos, etc., as I read through (and there are doubtless some that I have missed). I encourage the author to carefully check the galleys following PeerJ copyediting and layout. In particular, take please take a good look through the reference section, as I noticed a number of issues there.

For instance:

Line 129: missing parenthesis.

Line 215-216: confusing sentence

Line 217: should be “assumed”

Line 235: who/what is “they”?

Line 253: Acorn references, check title (needs colon)

Line 256: “hypothesis” should not be capitalized

Line 342: Italicize “T” in “The”, also “Toronto” not “Toronton”

Thank you for your contribution to PeerJ.

Reviewer 1 ·

Basic reporting

The revised manuscript and supporting documentation satisfied the questions I raised in the original version.

Experimental design

The author addressed my concern regarding the appropriate life stage to be sampled for evaluation of the ERH

Validity of the findings

The data supports the findings

·

Basic reporting

The flow of the paper has been improved.

Experimental design

The questions regarding the experimental design have been addressed.

Validity of the findings

The conclusions are appropriate.

Additional comments

This was a conscientious revision that addressed all my concerns.